# Structural studies reveal flexible roof of active site responsible for ω-transaminase CrmG overcoming by-product inhibition

Jinxin Xu[1,7 ✉], Xiaowen Tang[2,7], Yiguang Zhu[3], Zhijun Yu[1], Kai Su[1], Yulong Zhang[1,4], Yan Dong[1], Weiming Zhu[5], Changsheng Zhang [3], Ruibo Wu[2] & Jinsong Liu [1,4,6 ✉]

Amine compounds biosynthesis using ω-transaminases has received considerable attention in the pharmaceutical industry. However, the application of ω-transaminases was hampered by the fundamental challenge of severe by-product inhibition. Here, we report that ω-transaminase CrmG from *Actinoalloteichus cyanogriseus WH1-2216-6* is insensitive to inhibition from by-product α-ketoglutarate or pyruvate. Combined with structural and QM/MM studies, we establish the detailed catalytic mechanism for CrmG. Our structural and biochemical studies reveal that the roof of the active site in PMP-bound CrmG is flexible, which will facilitate the PMP or by-product to dissociate from PMP-bound CrmG. Our results also show that amino acceptor caerulomycin M (CRM M), but not α-ketoglutarate or pyruvate, can form strong interactions with the roof of the active site in PMP-bound CrmG. Based on our results, we propose that the flexible roof of the active site in PMP-bound CrmG may facilitate CrmG to overcome inhibition from the by-product.

[1] State Key Laboratory of Respiratory Disease, Guangzhou Institutes of Biomedicine and Health, Chinese Academy of Sciences, 510530 Guangzhou, China. [2] School of Pharmaceutical Sciences, Sun Yat-sen University, 510006 Guangzhou, China. [3] Key Laboratory of Tropical Marine Bio-resources and Ecology, Guangdong Key Laboratory of Marine Materia Medica, RNAM Center for Marine Microbiology, South China Sea Institute of Oceanology, Chinese Academy of Sciences, 164 West Xingang Road, 510301 Guangzhou, China. [4] Graduate University of Chinese Academy of Sciences, 100049 Beijing, China. [5] Key Laboratory of Marine Drugs, Ministry of Education of China, School of Medicine and Pharmacy, Ocean University of China, 266003 Qingdao, China. [6] Guangdong Provincial Key Laboratory of Biocomputing, Guangzhou Institutes of Biomedicine and Health, Chinese Academy of Sciences, 510530 Guangzhou, China. [7] These authors contributed equally: Jinxin Xu, Xiaowen Tang. ✉email: xu_jinxin@gibh.ac.cn; liu_jinsong@gibh.ac.cn

Amine moieties are recognized as one of the vital chemical building blocks forming bioactive natural products, agrochemicals, as well as pharmaceutical drugs[1,2]. The development of effective and broadly applicable methods for amine compound production has been regarded as a research priority in the pharmaceutical industry. Chemical transformation is currently the predominantly used approach for chiral amines synthesis. However, the drawbacks of the conventional chemical method are apparent, including low enantiopurity selectivity, environmental harm, and sometimes the use of precious and even toxic metal catalysts[2]. In virtue of environmental friendliness, external cofactor independence, extraordinary enantioselectivity and regioselectivity, ω-transaminases have gained considerable attractions in chiral amine synthesis[1]. ω-transaminases are a subgroup of pyridoxal 5′-phosphate (PLP)-dependent enzymes, which are capable of catalyzing amination of keto acids, aldehydes, and ketones. This catalytic process consists of two half-reactions[2,3]. In the first half-reaction, PLP accepts an amino group from amine (amino donor), resulting in the generation of pyridoxamine 5′-phosphate (PMP)-bound enzyme and ketone or aldehyde by-product; in the second half-reaction, the amino group is transferred to an amino acceptor (keto acid, aldehyde, or ketone), leading to new amine production and PLP regeneration.

Despite the tremendous advantage and potential, the widespread application of ω-transaminases for organic synthesis is facing the fundamental challenges of severe by-product inhibition and unfavorable equilibrium[1,4–7]. To overcome these challenges, several by-product removal strategies have been developed. Two coupled enzymatic systems were used to convert pyruvate, the corresponding by-product of universal amino donor L-Ala that is accepted by almost all transaminases[8,9], including L-Ala dehydrogenase/glucose dehydrogenase system that converts pyruvate to L-Ala[10], and lactate dehydrogenase/glucose dehydrogenase system that converts pyruvate to lactate[1,4,11,12]. However, the large-scale application of this strategy was limited by the high costs of the coupled enzyme system. Using isopropylamine as an amino donor, the corresponding by-product acetone can be removed by evaporation under reduced pressure[1,4,13,14]. Unfortunately, isopropylamine is not widely accepted for ω-transaminases[9].

Aside from by-product removal, protein engineering was developed to overcome the by-product inhibition of ω-transaminase. In 2005, through random mutagenesis, Yun et al.[15] developed a mutant of ω-transaminase from *Vibrio fluvialis JS17* with reduced by-product inhibitory effect. However, it is still difficult to obtain ω-transaminase completely lacking by-product inhibition through enzyme engineering. Interestingly, in 2013, Park et al.[16] reported that ω-transaminase from *Ochrobactrum anthropi* (OA-ωTA) is devoid of inhibition from by-product acetophenone in the transamination between amino acceptor pyruvate and amino donor (S)-α-MBA. However, the mechanism of how OA-ωTA eliminates inhibition from the ketone product is still unknown. Moreover, until now, no ω-transaminase has been reported to circumvent the inhibition from by-product pyruvate, which is derived from the deamination of the universal amino donor L-Ala. Thus, it is highly desirable to discover by-product inhibition-free ω-transaminases and understand the mechanism, which can efficiently facilitate the rational design of ω-transaminases completely devoid of by-product inhibition and eventually eliminate the obstacle for widespread application of ω-transaminases.

Previously, the biosynthetic pathway of the immunosuppressive agent caerulomycin A (CRM A) from the marine actinomycete *Actinoalloteichus cyanogriseus WH1-2216-6* has been identified[17–20], and the critical role of the (S)-selective ω-transaminase CrmG in the biosynthesis of CRM A has been reported. CrmG catalyzes amino acceptor caerulomycin M (CRM M) transformation to caerulomycin N (CRM N)[21]. In this study, we report that, without by-product removal in situ, high conversion of CRM M can be achieved by CrmG with amino donor L-Ala, L-Gln, or L-Glu. Moreover, the activity of CrmG is insensitive to inhibition from ketone α-ketoglutarate or pyruvate. To uncover the mechanism on how CrmG eliminates by-product inhibition, we determine the crystal structures of the apo form CrmG, as well as its complexes with amino donor L-Ala, L-Gln, or L-Glu. Combined with detailed structural analysis of CrmG in complex with the cofactor (PLP or PMP), or amino acceptor CRM M, we not only establish the reaction mechanism of CrmG-catalyzed transamination, but also, more importantly, elucidate the mechanism of CrmG overcoming by-product inhibition.

## Results

**ω-Transaminase CrmG is insensitive to by-product inhibition**. Severe by-product inhibition is a common feature of transaminase[1,4,5,7]. *Vibrio fluvialis JS17* transaminase (Vf-ωTA) was strongly inhibited by the by-product 2-butanone with a $K_i$ of 5.1 mM[15]; the activity of Vf-ωTA catalyzing acetophenone amination was completely lost in the presence of 5 mM pyruvate[11]. Previously, we reported that ω-transaminase CrmG can effectively catalyze CRM M amination with some amino acids, especially L-Glu, as amino donors[21] (Fig. 1a). To determine the susceptibility of CrmG to the by-product inhibition, here, we determined the inhibition constant of α-ketoglutarate, which is the by-product in L-Glu deamination, against CrmG-catalyzed reaction. Interestingly, we found that the inhibitory activity of α-ketoglutarate toward CrmG is very weak with a $K_i$ value of 145.9 mM (Fig. 1b). In addition, CrmG is also tolerant of the inhibition from by-product pyruvate with a $K_i$ value of 225.9 mM (Fig. 1c).

**Favorable equilibrium of CRM M conversion by CrmG**. Besides by-product inhibition, unfavorable equilibrium caused by by-product is another fundamental hurdle for the widespread application of transaminase[1,4,8]. To determine whether CRM M conversion suffers from unfavorable equilibrium, we investigate the equilibrium of CrmG catalyzing CRM M conversion. We previously reported that, without by-product removal in situ, biaryl aldehyde CRM M can be completely converted to biaryl amine CRM N by CrmG using high excess (50 equiv) of L-Glu, L-Gln, or L-Ala as the amino donor (Supplementary Table 1)[21]. Here, we carried out reactions of CrmG catalyzing CRM M amination with 1.5 equiv amino donor L-Glu, L-Gln, or L-Ala (Fig. 1d). Our results showed that, without by-product removal in situ, about 90% CRM M can be converted to CRM N with amino donor L-Glu and L-Gln, in 12 and 48 h, respectively. With amino donor L-Ala, 50% conversion of CRM M to CRM N was achieved in 48 h. This indicates that high conversion of CRM can be achieved with little excess amino donor L-Glu, L-Gln, or L-Ala.

**Crystal structure of apo form and amino donor-bound form CrmG**. To illustrate the mechanism of CrmG overcoming by-product inhibition, here, we determined the structures for apo and amino donor-bound form CrmG. The crystal structures of the apo form CrmG were determined to resolution 2.10 and 2.85 Å in space group I222 and C2, with one and two dimers in the asymmetric unit, respectively. The crystal structures of CrmG in complex with amino donor L-Gln, L-Glu, or L-Ala in P1 space group were determined to resolution 2.35, 2.25, and 2.20 Å, respectively. For these three crystal structures, each asymmetric unit contains two dimers.

To confirm that the dimeric CrmG form shown in the crystal structure is not due to crystal packing artifact, here, we performed

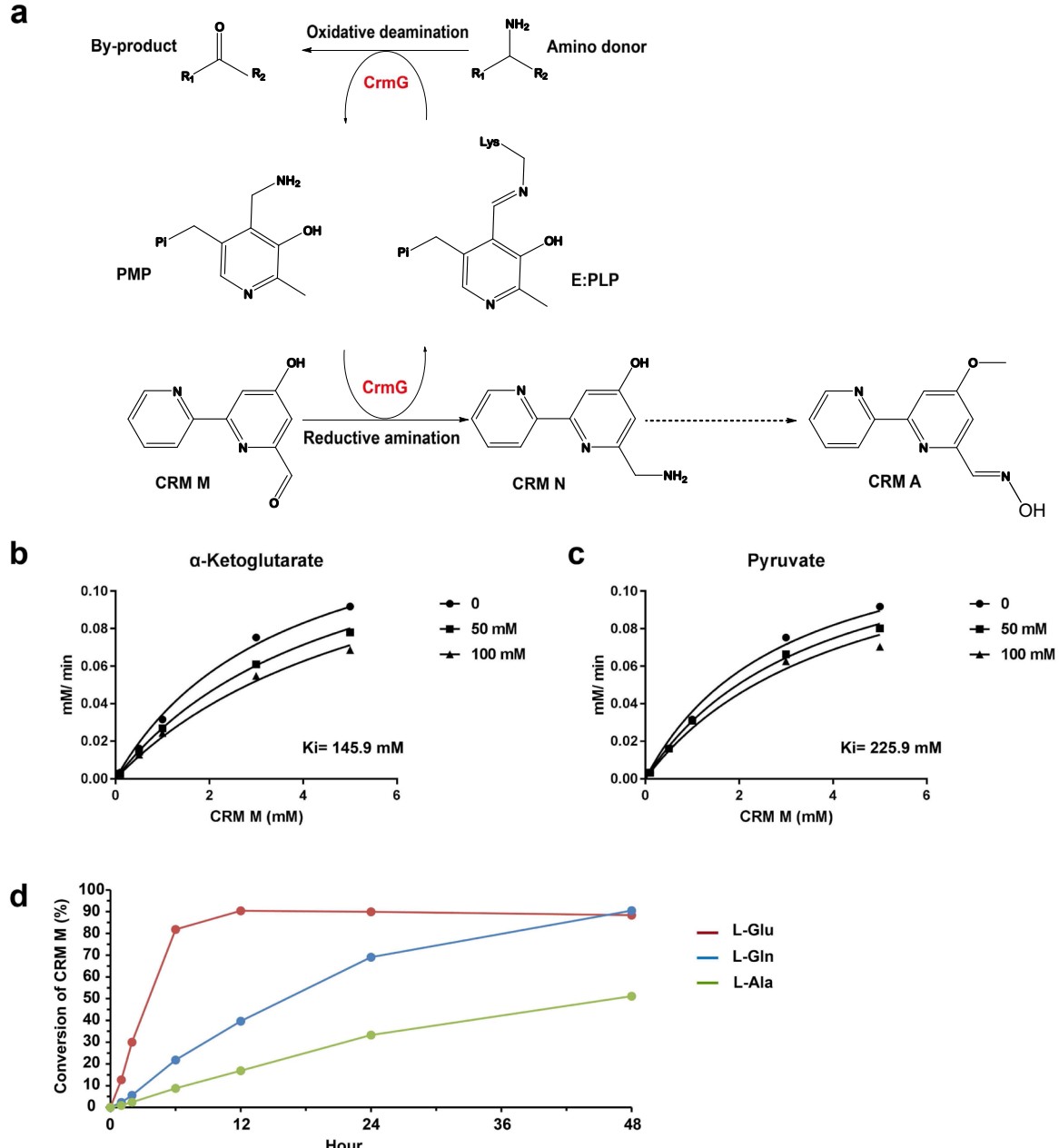

**Fig. 1 Catalytic characterizations of CrmG. a** Scheme for CrmG-catalyzed reaction. CrmG reaction pathway consists of oxidative deamination of an amino donor, resulting in by-product and PMP formation, and reductive amination of amino acceptor CRM M, leading to CRM N production, and PLP regeneration. **b** Nonlinear regression analysis of the competitive inhibition of α-ketoglutarate. **c** Nonlinear regression analysis of the competitive inhibition of pyruvate. **d** CrmG reaction progress with 1.5 equivalent amino donor. The reaction was performed in 50 μL assay solution (50 mM Tris pH7.5, 0.1 M NaCl, and 5% Glycerol) containing 4 μM CrmG, 80 μM PLP, 0.3 mM amino donor (L-Glu, L-Gln, or L-Ala), and 0.2 mM CRM M, in triplicates.

the solution study on apo CrmG using Dynamic light scattering (DLS). The calculated molecule weight of CrmG in solution is 114 kDa (Supplementary Fig. 1), whereas the theoretical molecule weight of CrmG is 59.22 kDa. DLS results suggested that CrmG can form a dimer in solution.

**PLP binding induces the rearrangement and stabilization of CrmG active site.** We have previously defined the catalytic pocket of CrmG as located on the dimer interface, containing cofactor (PLP or PMP) binding site and substrate binding site separated by catalytic base K344[21]. In the active form CrmG (PLP or PMP-bound CrmG), β5/α6 loop and α6-α7 helices cover on top of the active site, hereafter we named this region "roof" (Fig. 2a).

However, the catalytic pocket in apo CrmG is structurally different from that in the active form. In the structure of apo CrmG in C2 space group, we observe four similar catalytic pockets in two CrmG dimers. In these structures, the "roof" region, i.e., the segment between β5 and β6 from one monomer and α11′ from another monomer is disordered or flexible (Figs. 2b, 3a; Supplementary Fig. 2). The structural difference of the "roof" in active form (Supplementary Fig. 3a) and apo form (Supplementary Fig. 3b, c) is also validated by molecular dynamics (MD) simulations, and root-mean square fluctuation (RMSF) (Supplementary Fig. 3d) indicates that the "roof" in apo CrmG of C2 space group is more flexible. Interestingly, in the structure of apo CrmG in I222 space group, the two catalytic pockets of CrmG dimer show completely different conformations. One catalytic pocket

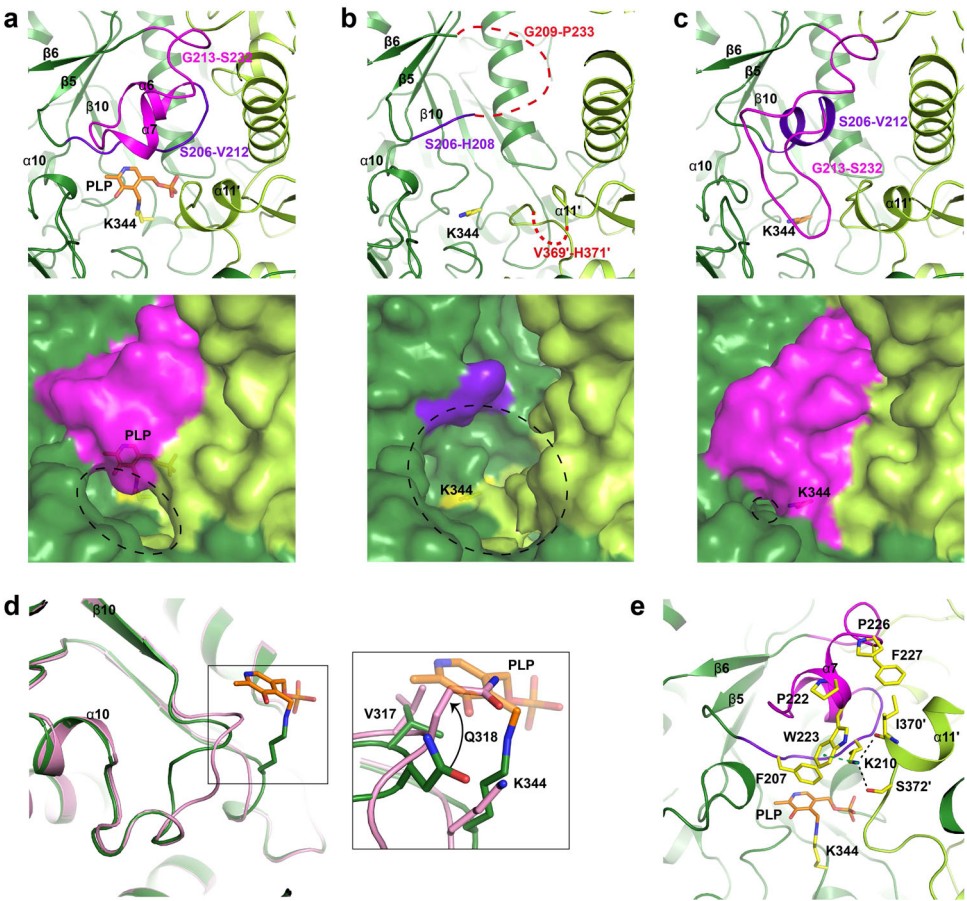

**Fig. 2 Conformation changes on the active site of PLP-bound CrmG and apo CrmG. a** Structure of the active site in PLP-bound CrmG (PDB code 5DDS). The structure is presented as cartoon (upper panel) and surface (lower panel). The active site entrance or opening is labeled with dash circle. Monomer 1 is shown in deep green and monomer 2 is shown in light green. PLP is shown as orange stick. Catalytic base K344 is shown as yellow stick. Residues S206-V212 colored purple blue, residues G213-S232 colored magenta. **b** One of the four active sites of apo CrmG in C2 space group. Residues S206-H208 colored purple blue, disordered residues G209-P233 in monomer 1 and V369'-H371' in monomer 2 are shown as red dash line. **c** The closed form active site of apo CrmG in I222 space group. Coloring scheme same as in **a**. **d** Comparison of β10/α10 loop and K344 on apo CrmG (pink) with that on PLP-bound CrmG (deep green). **e** PLP induces the rearrangement and stabilization of active site in PLP-bound CrmG. Cation–π interaction is indicated with green dash line.

adopts the conformation similar with that of apo CrmG in C2 space group, in which the catalytic base K344 is exposed to solvent, here we name it "open" pocket; whereas the other catalytic pocket exhibits closed conformation, in which the G213-S232 loop acts like a lid covering on top of the catalytic K344, therefore K344 becomes inaccessible to the solvent (Fig. 2c). The closed pocket is more ordered than the open pocket (Fig. 2b, c and Fig. 3a, b). Residue W223 from the closed pocket forms a cation–π interaction with R451 from a symmetry-related molecule (Supplementary Fig. 4), indicating that the closed pocket might be stabilized by crystal packing. In addition, in the catalytic pocket of the closed apo form, residues S206-V212 form a helix, whereas this segment is a loop in both open apo form and cofactor-bound active form (Fig. 2a–c; Supplementary Fig. 3a–c). Another interesting difference between apo and active catalytic pocket is the conformation of V317-I321 around the β10/α10 loop. In the active pocket, the aromatic ring of PLP sits on V317. However, in both open and closed apo pockets, the V317-I321 region adopts a more relaxed conformation, and Q318 occupies the position of the aromatic ring of PLP (Fig. 2d). Structure analysis and key distances tracking along the MD trajectory also validate the position shift of Q318. As shown in Supplementary Fig. 5, the distance between side chains of V317 and Q318 is more fluctuant in the apo from. Moreover, the main chain shift of

catalytic base K344 is also observed between apo and active form CrmG (Fig. 2d). This is confirmed by the MD analysis that reveals more fluctuating and longer distances between V317 and K344 (both main chain and side chain) in apo form (Supplementary Fig. 5). These indicate that the catalytic pocket in apo CrmG is dynamic, and PLP binding induces the structural rearrangement.

To further elucidate how PLP binding induces the structural rearrangement of the catalytic pocket, we revisit the crystal structure of PLP-bound form CrmG (PDB code 5DDS)[21]. As shown in Fig. 2e, there are two Proline residues (P222 and P226) located on the first and second turn of α7 helix, which is part of the "roof" region. These two proline residues, leading to the loss of backbone hydrogen bonds, potentially destabilize the α7 helix and contribute to the flexibility of the roof. When PLP is bound, it can induce and stabilize the S206-V212 region to adopt appropriate conformation via interaction with F207, hence enabling K210 to interact with the side chain of S372' and the main chain of I370' from the adjacent monomer via hydrogen bonds. The interactions of K210 with S372' and I370' further stabilize the conformation of α11' helix (residues P365'-I370'). The F207 and K210 also interact with W223 on α7 helix through hydrophobic interaction and cation–π stack, respectively; and I370' on α11' helix forms hydrophobic interactions with P222, W223, P226, and F227 on α7 helix. Therefore, the interactions

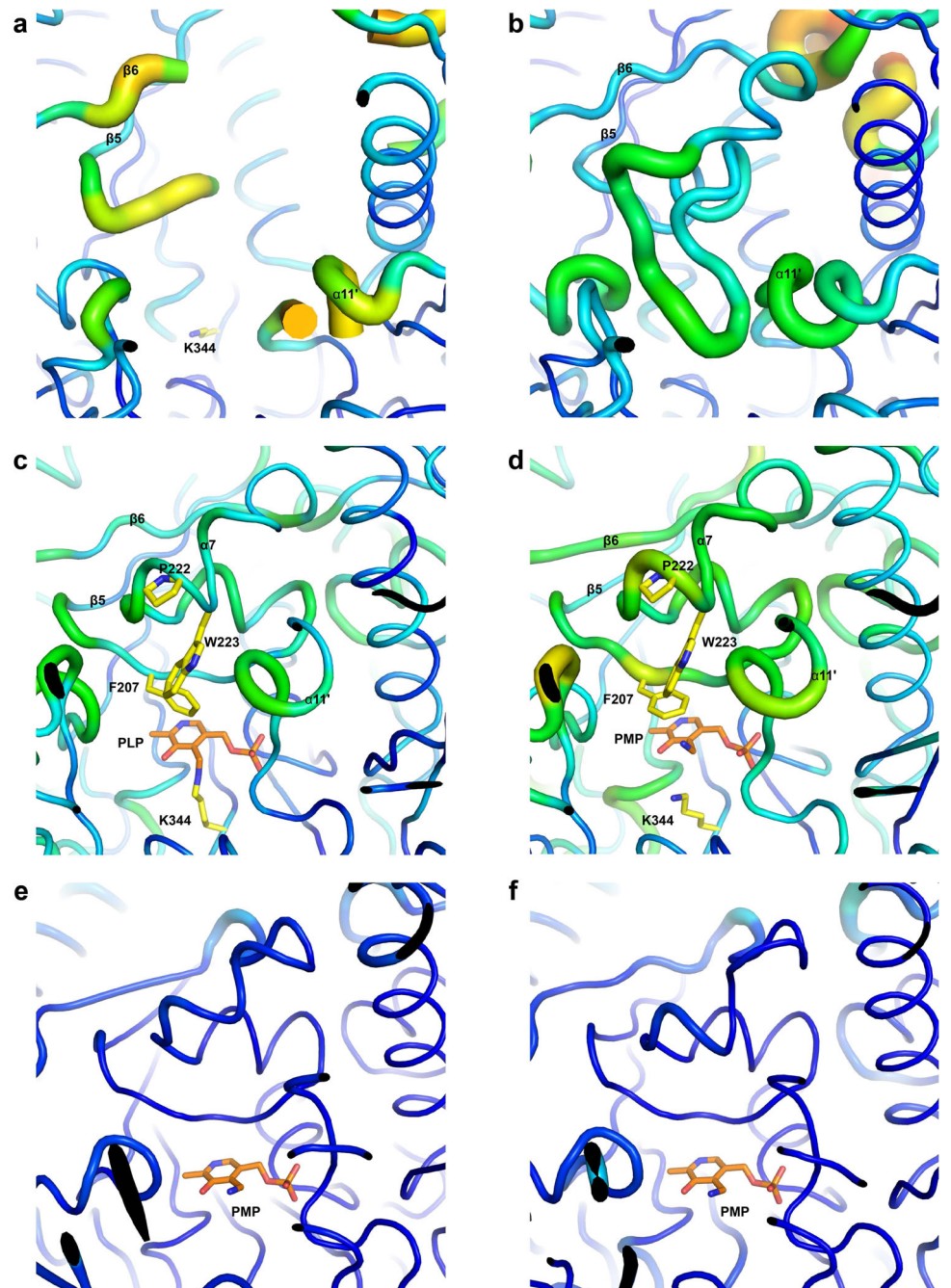

**Fig. 3 B-factor putty representation of the catalytic pocket for apo and active form CrmG. a** B-factor putty representation of the open form catalytic pocket of apo CrmG in C2 space group. **b** B-factor putty representation of the closed form catalytic pocket of apo CrmG in I222 space group. **c** B-factor putty representation of the activated catalytic pocket of PLP-bound CrmG (PDB code 5DDS). **d** B-factor putty representation of the activated catalytic pocket of PMP-bound CrmG (PDB code 5DDU). **e** B-factor putty representation of the activated catalytic pocket of PMP-bound Vf-ωTA (PDB code 4E3Q). **f** B-factor putty representation of the activated catalytic pocket of PMP-bound OA-ωTA (PDB code 5GHF). The cartoon thickness and color reflect the relative Cα B-factors within the molecule. PLP or PMP is shown as orange stick.

with F207 and K210 and α11′ helix are essential for the formation and stability of α7 helix. Hence, the structural rearrangement of the "roof" is induced by PLP.

To our knowledge, although numerous structures of ω-transaminase have been determined, here, we firstly report a closed form catalytic pocket of apo ω-transaminase with the active base inaccessible to the solvent. Nevertheless, the disordered roof of the active site has been reported in other apo form ω-transaminases. In apo form ω-transaminase from *Chromobacterium violaceum* (Cv-ωTA), the flexibility of the roof

depends on crystallization condition[22]. This region is disordered in the apo form Cv-ωTA crystallized with PEG3350, but it folds into the active conformation when crystallized from a solution containing poly(acrylic acid). Besides CrmG and Cv-ωTA, the disordered roof region is also observed in one molecule of the apo form structure of catabolic *N*-succinylornithine transaminase (AstC)[23]. However, the roof in the other molecule adopts active conformation in the apo AstC structure. This suggests that a certain degree of flexibility in the roof may be a general property of apo ω-transaminase. This flexibility will facilitate the PLP entry

to the active site, as PLP sits deeply in the catalytic pocket and is covered by the roof region of ω-transaminase.

Other than the roof of the active site, the structural rearrangement of the interfacial loop, outer loop, and N terminal on the active site cavity was also observed in Cv-ωTA upon PLP binding[22,24]. A more recent study proposed that the structural change of the later element is triggered by a steric clash between the free catalytic lysine and Y322′ on the interfacial loop, which is induced by the conformational change of catalytic lysine in apo Cv-ωTA[24]. Moreover, this clash was also proposed to promote the dissociation of apo Cv-ωTA dimer. However, in CrmG, the Y322 is replaced by F375, and the free catalytic lysine moves away from F375′ in the apo CrmG (Supplementary Fig. 6). Thus, in apo CrmG, the free catalytic lysine is unable to induce the structural rearrangement of the interfacial loop and the dimer dissociation.

**Flexibility of the active site in PMP-bound CrmG.** As mentioned above, structural rearrangement and stabilization of the roof in the active site of CrmG depend on PLP binding, which is covalently linked to the catalytic base residue K344. In the first half-reaction of a typical transaminase reaction (amino donor deamination), PLP is converted into PMP and then released from K344. This indicates that the interaction of PMP with CrmG is weaker than that of PLP, which potentially leads to the increased flexibility of the roof region in PMP-bound CrmG. Crystallography temperature factor (B-factor) distribution analysis clearly shows that active site in PMP-bound CrmG is less ordered than that in the PLP-bound CrmG (Fig. 3c, d). The flexibility of the active site roof region in PMP-bound CrmG is increased, compared with that in PLP-bound CrmG, especially for residues F207 and P222. For comparison, we further analyze the B-factor distribution for the structures of PMP-bound form ω-transaminase from *V. fluvialis* (Vf-ωTA) (PDB code 4E3Q), PMP-bound form ω-transaminase from *O. anthropi* (OA-ωTA) (PDB code 5GHF). Different from CrmG, both Vf-ωTA and OA-ωTA prefer pyruvate as amino acceptor[14,25]. The B-factor distribution analysis clearly shows that the roof region remains well ordered in both PMP-bound Vf-ωTA and OA-ωTA structures (Fig. 3e, f).

In 2009, Schell et al.[26], used transaminase to synthesize PMP, indicating that PMP can be released from the active site of some transaminases. We then hypothesized that the PMP can also be easily released from CrmG as a result of the increased flexibility of the roof at the active site in PMP-bound CrmG. To test this notion, we performed the PLP conversion reaction (0.02 mM CrmG, 0.2 mM PLP, and 2 mM amino donor). As shown in Fig. 4a, the time course of UV–vis spectral changes reveals the transamination reaction catalyzed by CrmG with PLP (absorption $\lambda_{max} \approx 414$ nm) converted to PMP (absorption $\lambda_{max} \approx 323$ nm). Within 48 h, almost all PLP is converted to PMP using excess amino donor L-Glu, L-Gln, or L-Ala. We should note that the concentration of PLP is 10 times of CrmG in these reactions. This indicates that the synthesized PMP should be released from CrmG, which then allows new PLP to bind to the active site and be converted to PMP. Our structures show that PMP/PLP is deeply buried in the catalytic pocket and covered by the roof (Figs. 2a and 3c, d). For PMP to be released form CrmG, the roof region of the active site on PMP-bound CrmG must have a certain degree of flexibility. Therefore, we speculated that increasing the flexibility of the active site roof would accelerate the displacement of PMP by free PLP. We then mutated W223 on α7 helix to alanine, as W223 can stabilize α7 helix by interacting with F207 and K210 and I370′. The efficiency of CrmG catalyzing PLP conversion to PMP is greatly improved by the mutation W223A (Fig. 4a, b). Taken together, these results suggest that the

active site of PMP-bound form CrmG is destabilized comparing to that of PLP-bound CrmG.

**Structure basis of amino donor recognition by CrmG.** To further understand the amino donor deamination reaction of CrmG, here we analyze the crystal structures of CrmG with different amino donors: L-Gln, L-Glu, and L-Ala. In the crystal structure of CrmG with L-Gln, only one of four active sites holds L-Gln. The L-Gln does not form external aldimine with PLP as no continuous electron density can be observed between the N atom of L-Gln and C4′ atom of PLP (Supplementary Fig. 7a). The L-Gln is anchored in the substrate cavity via interactions with residues from both CrmG monomers (Fig. 5a). The L-Gln is sandwiched between F55 and F207. The α-carboxylate of L-Gln forms a salt bridge with R486. The side chain of L-Gln orients toward the second monomer and the γ-carboxamide group interacts with S372′ through a hydrogen bond. Most interestingly, the PLP also contributes to the L-Gln capture by forming a hydrogen bond with N atom of L-Gln via its phenolic OH group. With these interactions, the L-Gln should be held tightly in the substrate cavity. We also notice that the distance between N atom of L-Gln and C4′ of PLP is 3.1 Å, suggesting that the L-Gln is ready to react with PLP to form the external aldimine.

In the structures of CrmG with L-Glu or L-Ala, clear electron density was observed between the N atoms of those amino donors and the C4′ atom of PLP (Supplementary Fig. 7b, c), indicating that we have captured the external aldimine for these amino acids. In the structure of CrmG with PLP-Glu, γ-carboxylate of the L-Glu portion not only forms a hydrogen bond with S372′ but also forms an additional salt bridge with K210 (Fig. 5b). In the structure of CrmG with PLP-Ala, due to the shorter side chain and lack of γ-carboxylate or γ-carboxamide group, the L-Ala portion only forms contact with residues of R486 and Q318 from one monomer via salt bridge and hydrogen bond interaction, respectively, without interacting with the second monomer of CrmG dimer (Fig. 5c).

Based on these three crystal structures, amino donors L-Glu, L-Gln, and L-Ala exhibit strong, moderate, and weak binding to CrmG, respectively, due to side chain differences. These corroborate well with our kinetic data that, $K_m$ value for L-Glu, L-Gln, and L-Ala toward CrmG is 80.02, 182.70, and 405.10 μM, respectively (Fig. 5d).

**Energetic characteristics of the transamination reaction.** To deeply understand the equilibrium of CrmG catalyzing CRM M amination, we further calculated the reaction energy profile of CrmG, using a combined quantum mechanics/molecular mechanics (QM/MM) approach. The energy profile for CrmG catalyzing L-Glu deamination was calculated based on the crystal structures of PLP-bound CrmG and L-Glu-bound CrmG; the energy profile for CrmG catalyzing CRM M amination was calculated based on structures of PMP-bound CrmG and CRM M-bound CrmG. Our calculated results suggested that the deamination of L-Glu contains three stages, integration of amine to generate geminal diamine (Stage A, A-1 to A-4), dissociation of geminal diamine to generate external aldimine (Stage B, B-1 to B-3), and hydrolysis of external aldimine (Stage C, C-1 to C-3) (Fig. 6a). Similarly, the amination of CRM M can be regarded as a reverse process of the deamination process. For the reaction mechanism, our proposed three stages in the transamination process agree with that previously reported by Himo et al.[27,28].

As shown in Fig. 6b, CrmG catalyzing L-Glu deamination is an exothermic reaction with calculated reaction enthalpy of −33.5 kcal/mol, and CRM M amination is endothermic with calculated reaction enthalpy of 11.5 kcal/mol. The QM/MM results suggest

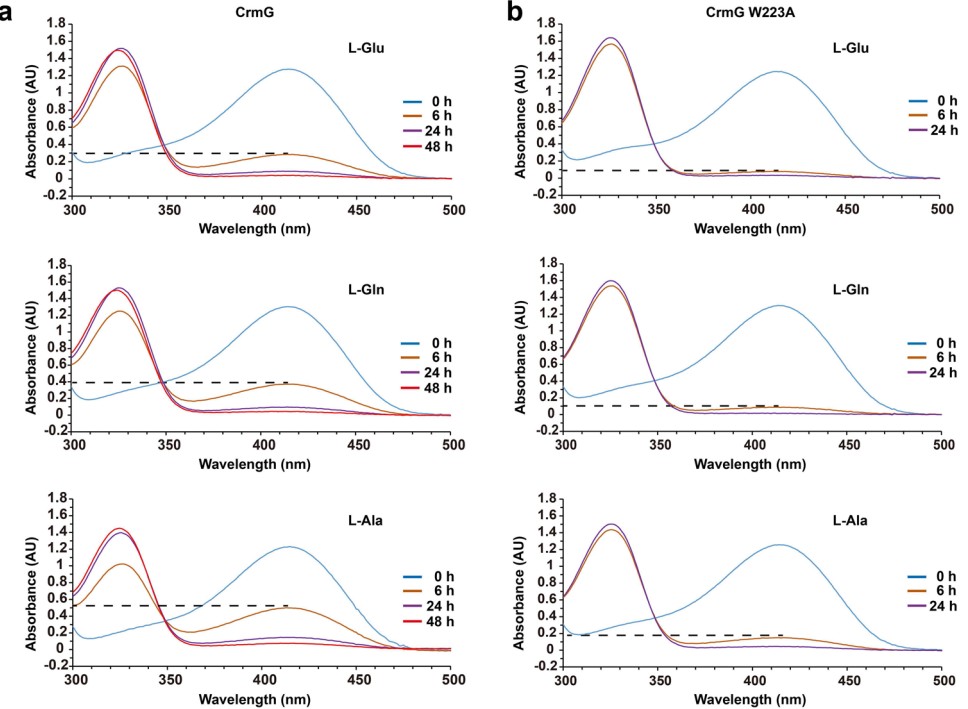

**Fig. 4 Analysis of CrmG catalyzing PLP amination with excess amino donor. a** UV–vis spectrophotometry analysis of CrmG catalyzing PLP conversion to PMP with excess amino donor (ʟ-Glu, ʟ-Gln, or ʟ-Ala). **b** UV–vis spectrophotometry analysis of CrmG W223A mutant catalyzing PLP conversion to PMP with excess amino donor (ʟ-Glu, ʟ-Gln, or ʟ-Ala). The assays were conducted in 1 mL of reaction mixture in Tris-HCl buffer (50 mM Tris pH 7.5, 0.1 M NaCl, 5% Glycerol), comprising of 0.02 mM CrmG or CrmG W223A mutant, 0.2 mM PLP, and 2 mM amino donor (ʟ-Glu, ʟ-Gln, or ʟ-Ala).

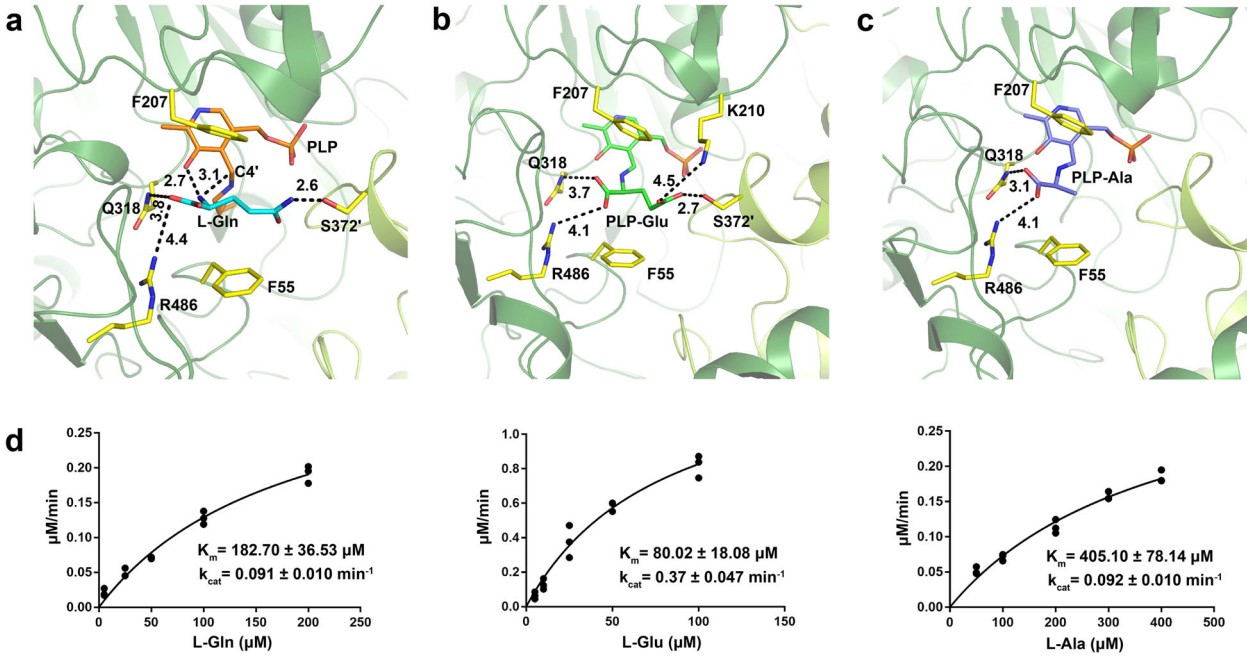

**Fig. 5 Interactions between CrmG and amino donors. a** Detailed interaction of ʟ-Gln with CrmG. ʟ-Gln was shown as cyan stick; internal aldimine was shown as orange stick, critical residues for ʟ-Gln interaction were shown as yellow stick. **b** Detailed interaction of ʟ-Glu with CrmG. PLP-Glu was shown as green sticks. **c** Detailed interaction of ʟ-Ala with CrmG. PLP-Ala was shown as light blue sticks. **d** Determination kinetic parameters of CrmG toward amino donor ʟ-Gln, ʟ-Glu, or ʟ-Ala, in triplicates.

that ʟ-Glu deamination is energetically more favorable than CRM N deamination, and CRM M amination is energetically more favorable than α-ketoglutarate amination. This implies that by-product α-ketoglutarate and product CRM N cannot induce severe unfavorable equilibrium for CrmG catalyzing CRM M amination with amino donor ʟ-Glu. This corroborates well with our biochemical results that high conversion of CRM M by CrmG can be achieved using ʟ-Glu as an amino donor.

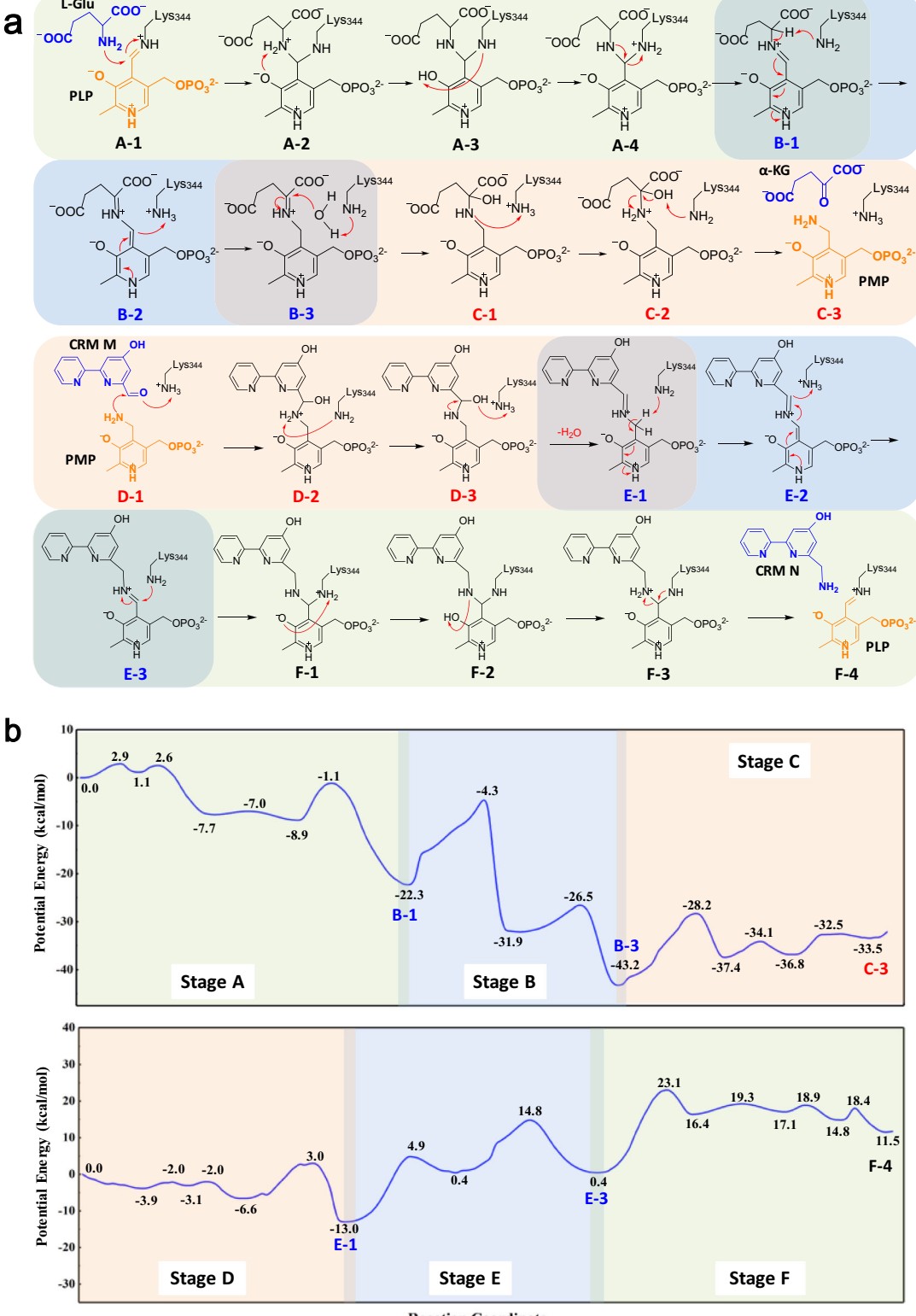

**Fig. 6 QM/MM study of CrmG reaction mechanism. a** Detailed reaction mechanism of CrmG-catalyzed transamination. Cofactors PLP and PMP are colored in orange; amino donor L-Glu, amino acceptor CRM M and by-product α-ketoglutarate (α-KG) are colored in blue. **b** Detailed energy profile for CrmG-catalyzed transamination. L-Glu and CRM M were used as amino donor and amino acceptor, respectively.

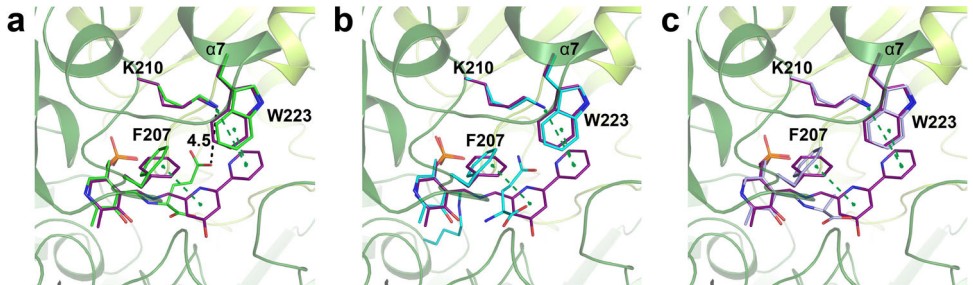

**Fig. 7 CRM M stabilizes the roof of the active site on CrmG. a** Comparison of the structures of CrmG in complex with CRM M (purple) and CrmG in complex with L-Glu (green). **b** Comparison of the structures of CrmG in complex with CRM M (purple) and CrmG in complex with L-Gln (cyan). **c** Comparison of the structures of CrmG in complex with CRM M (purple) and CrmG in complex with L-Ala (light blue). Pi interactions are indicated with green dash line.

**By-product inhibition elimination mechanism of CrmG.** It had been well demonstrated that pyruvate was a highly reactive amino acceptor for Vf-ωTA[25] and OA-ωTA[14], and the acetophenone amination by Vf-ωTA was severely inhibited by pyruvate[11]. In the docked structure of PMP-bound form OA-ωTA in complex with pyruvate, the carboxyl group of pyruvate was coordinated by hydrogen bonds with residue W58 and R417[29], which are conserved in Vf-ωTA. Similarly, in the structure of CrmG in complex with L-Ala (corresponding amino donor for by-product pyruvate), the carboxyl group of L-Ala forms a hydrogen bond with Q318 and a salt bridge with R486 (Fig. 5c). However, CrmG is insensitive to pyruvate inhibition with $K_i$ of 225.9 mM. Moreover, in the case of L-Glu, which forms stronger interactions with the active site of CrmG than L-Ala, the corresponding by-product α-ketoglutarate also exhibits very weak inhibitory potential with a $K_i$ value of 145.9 mM. Our structural analysis shows that the roof of the active site in PMP form CrmG is flexible, whereas this region is well ordered in both PMP form Vf-ωTA and OA-ωTA (Fig. 3d–f). Our biochemical results also indicate that the flexible roof of the active site can facilitate PMP releasing from CrmG. This implies that the flexible roof of the active site can also promote by-product dissociation from CrmG. Thus, our results suggest that the flexibility of the active site will protect CrmG from by-product inhibition.

Based on the above-mentioned analysis, for effective binding to the active site, the amino acceptor should provide enough energy to stabilize the roof of the active site in PMP-bound form CrmG. Previously, we reported the crystal structure of CrmG with amino acceptor CRM M[21]. Compared to L-Glu/L-Gln/L-Ala, CRM M not only forms interaction with F55, and with R486 and S372′ via water molecules, but also forms strong interactions with the roof of CrmG through π–π interactions with F207 and W223 and cation–π interaction with K210 (Fig. 7a–c; Supplementary Fig. 8). This indicates that the roof of the active site on PMP-bound CrmG can be effectively stabilized by amino acceptor CRM M. We also showed that amination activity toward CRM M was reduced for CrmG W223A and K210E mutants with amino donor L-Gln[21]. W223 and K210 are not required for CrmG binding with L-Gln but involved in the interaction with CRM M and the stabilization of the roof on CrmG active site. This strongly implies that stabilizing the roof of CrmG is essential for CRM M amination.

Taken together, our results indicate that stabilizing the active site in PMP-bound CrmG is essential for amino acceptor amination and by-product inhibition. Thus, we propose a mechanism for the by-product inhibition elimination for CrmG (Supplementary Fig. 9). In apo CrmG, the active site is dynamic, which might switch between open and closed forms. When apo CrmG is in its open form, the PLP binds to the active site, resulting in active site stabilization. Then the deamination reaction occurs. The amino group is transferred from the amino

donor (L-Ala, L-Glu, or L-Gln) to PLP, generating by-product and PMP. In PMP-bound CrmG, the active site is dynamic and flexible. The by-products converted from L-Ala/L-Glu/L-Gln deamination are easily released from the PMP-bound form CrmG, owing to the flexible roof of the active site and the inability of the by-products to stabilize the flexible roof. Hence, the by-product cannot prevent CRM M from binding to the active site of PMP-bound form CrmG. Furthermore, CRM M binding can stabilize the roof of the active site in the PMP-bound CrmG, which can enable CRM M to form intermediate with PMP, resulting in CRM N formation and PLP regeneration.

## Discussion

Severe by-product inhibition is the fundamental challenge for the widespread application of ω-transaminase to synthesize amine compounds. Here, we identified that ω-transaminase CrmG, which is involved in the biosynthetic pathway of the immunosuppressive agent CRM A, is insensitive to the inhibition from the by-product. Combined with biochemical, structural, and QM/MM studies, we provide the detailed catalytic mechanism of CrmG. Particularly, we gain structural insights on how CrmG overcomes inhibition from the by-product. Further studies, especially on protein engineering, will need to be carried out to validate our proposed mechanism of overcoming by-product inhibition of ω-transaminases. Thus, our results not only shed light on how CrmG circumvents the by-product inhibition, but also may facilitate the rational design of transaminase to eliminate by-product inhibition.

## Methods

**Protein expression and purification.** CrmG was expressed and purified as described previously[21]. Briefly, the cDNA encoding CrmG was inserted into pET28a vector. CrmG was expressed in *E. coli* strain BL21 at 16 °C overnight and induced with 0.3 mM IPTG. The protein has an N-terminal His-tag and additional 14 residues from the vector. Purification of CrmG was performed using Ni affinity chromatography and followed by gel filtration. For crystallization, CrmG was concentrated to 10 mg/mL and dissolved in a buffer containing 20 mM Tris pH 7.5, 200 mM NaCl. For enzyme assay, CrmG was dissolved in a buffer containing 50 mM Tris pH 7.5, 100 mM NaCl and 5% glycerol.

**Dynamic light scattering experiments.** The CrmG with concentration of 4.8 mg/mL is dissolved in 50 mM Tris buffer pH 7.5. The dynamic light scattering experiments were performed at 20 °C using DynaPro Titan instrument.

**Crystallization.** All crystals of CrmG were grown at 293 K with sitting drop vapor diffusion method using the sparse matrix crystallization kit from Hampton research. The apo CrmG crystal in space group I222 was crystallized with 2 μL protein and 2 μL reservoir (30% PEG5000 MME, 100 mM Tris pH 8.0 and 200 mM Lithium sulfate); the apo CrmG crystal in space group C2 was crystallized with 4 μL protein (containing 2 mM PLP and 2 mM L-Gln) and 2 μL reservoir (0.2 M sodium acetate, 0.1 M Tris pH 8.5, 30% PEG 3350, 2% glycerol). To obtain crystals of L-Gln-bound, L-Glu-bound, and L-Ala-bound CrmG, CrmG was incubated with 5 mM PLP and 5 mM L-Gln, 5 mM PLP, and 5 mM L-Glu, 5 mM PLP, and 5 mM L-Ala prior to

**Table 1 Data collection and refinement statistics for CrmG structures.**

| | Apo1 | Apo2 | L-Gln | L-Glu | L-Ala |
|---|---|---|---|---|---|
| Data collection | | | | | |
| Space group | C2 | I222 | P1 | P1 | P1 |
| Cell dimensions | | | | | |
| $a$, $b$, $c$ (Å) | 122.13, 114.83, 156.17 | 115.09, 125.22, 155.38 | 84.04, 83.88, 88.41 | 84.09, 83.93, 88.46 | 83.76, 83.65, 87.97 |
| $\alpha$, $\beta$, $\gamma$ (°) | 90.00, 92.16, 90.00 | 90.00, 90.00, 90.00 | 106.48, 109.17, 94.98 | 106.60, 109.11, 95.13 | 106.48, 109.07, 95.12 |
| Resolution (Å) | 38.87–2.85 (2.94–2.85)[a] | 62.61–2.10 (2.15–2.10) | 50.58–2.35 (2.39–2.35) | 49.46–2.25 (2.29–2.25) | 78.58–2.20 (2.24–2.20) |
| Total observations | 117,939 (11,372) | 273,298 (20,018) | 267,360 (11,160) | 380,457 (18,981) | 350,386 (16,581) |
| Unique reflection | 47,035 (4432) | 64,955 (4542) | 81,461 (3461) | 96,761 (4822) | 99,929 (4846) |
| $R_{merge}$[b] | 0.110 (0.417) | 0.090 (0.634) | 0.082 (0.424) | 0.101 (0.513) | 0.072 (0.394) |
| $I/\sigma$ | 6.1 (2.1) | 11.1 (2.0) | 9.0 (2.7) | 8.6 (2.3) | 5.9 (2.0) |
| Completeness (%) | 93.5 (95.9) | 99.2 (99.9) | 91.2 (72.4) | 95.0 (95.5) | 92.9 (92.6) |
| Redundancy | 2.5 (2.6) | 4.2 (4.4) | 3.3 (3.2) | 3.9 (3.9) | 3.5 (3.4) |
| Refinement | | | | | |
| Resolution (Å) | 18.00–2.85 | 40.00–2.10 | 50.58–2.35 | 49.46–2.25 | 30.00–2.20 |
| No. of reflections | 44,504 | 61,786 | 77,461 | 92,048 | 94,926 |
| $R_{work}$[c]/$R_{free}$[d] (%) | 21.01 /26.18 | 18.59/22.83 | 17.17/20.75 | 17.69/21.46 | 20.36/25.42 |
| B-factors | | | | | |
| Protein | 40.084 | 34.012 | 36.208 | 31.777 | 41.623 |
| Ligand/ion | / | 55.907 | 37.803 | 36.952 | 45.546 |
| Water | 17.989 | 32.300 | 28.345 | 26.152 | 34.702 |
| R.m.s. deviations[e] | | | | | |
| Bond angles (°) | 1.5315 | 1.5379 | 1.5220 | 1.5844 | 1.6637 |
| Bond lengths (Å) | 0.0073 | 0.0083 | 0.0080 | 0.0087 | 0.0098 |
| Ramachandran plot[f] | | | | | |
| Most favored (%) | 93.41 | 96.33 | 96.79 | 96.31 | 95.94 |
| Allowed (%) | 5.81 | 2.98 | 2.22 | 2.71 | 3.61 |
| Outlier (%) | 0.78 | 0.69 | 0.99 | 0.98 | 0.45 |
| PDB no. | 6JCB | 6JCA | 6JC9 | 6JC8 | 6JC7 |

[a]The values in parentheses refer to statistics in the highest bin.
[b]$R_{merge} = \sum hkl \sum i|Ii(hkl) - <I(hkl)>|/\sum hkl \sum i Ii(hkl)$, where Ii(hkl) is the intensity of an observation and $<I(hkl)>$ is the mean value for its unique reflection; summations are over all reflections.
[c]$R$-factor $= \sum h|F_o(h) - F_c(h)|/\sum h F_o(h)$, where $F_o$ and $F_c$ are the observed and calculated structure-factor amplitudes, respectively.
[d]$R$-free was calculated with 5% of the data excluded from the refinement.
[e]Root-mean square-deviation from ideal values.
[f]Categories were defined by MolProbity.

crystallization, respectively. The crystals of amino donor-bound CrmG were grown in 0.2 M sodium acetate, 0.1 M Tris pH 8.5, 30% PEG 3350, and 2% glycerol.

**Data collection and structure determination**. All crystals were cryo-protected in mother liquid with addition of 20% glycerol, prior to collecting data at 100 K. Diffraction data for apo form, L-Glu-bound and L-Ala-bound CrmG crystals were collected at beam line 19U1 of the Shanghai Synchrotron Radiation Facility[30]. Diffraction data for L-Gln-bound CrmG crystal was collected at beam line 17U of the Shanghai Synchrotron Radiation Facility[31]. Data for apo CrmG, L-Glu-bound, and L-Gln-bound CrmG were processed with Mosflm[32], and data for L-Ala-bound CrmG was processed with DIALS[33]. All processed data were then scaled with Aimless[34]. All crystal structures were solved by molecule replacement with software Molrep[35] using PLP-bound CrmG structure (PDB code 5DDS) as the template. The refinement of the structures was conducted using Refmac5[36]. Data collection and refinement statistics are shown in Table 1. For apo CrmG structure in C2 space group, 486 residues, 483 residues, 487 residues, and 489 residues were modeled in chain A, B, C, and D, respectively. For apo CrmG structure in I222 space group, 505 residues and 514 residues were modeled in chain A and B, respectively. For structure of CrmG in complex with amino donor L-Gln, 512 residues, 512 residues, 512 residues, and 508 residues were modeled in chain A, B, C, and D, respectively. For structure of CrmG in complex with amino donor L-Glu, 512 residues were modeled in each of the four molecules in the asymmetric unit. For structure of CrmG in complex with amino donor L-Ala, 511 residues, 511 residues, 510 residues, and 512 residues were modeled in chain A, B, C, and D, respectively.

**CrmG enzyme assays**. For PLP conversion reaction by CrmG, the enzyme assays were carried out in 1 mL reaction mixture in Tris buffer (containing 50 mM Tris pH 7.5, 100 mM NaCl, 5% glycerol) by incubating 0.02 mM CrmG or CrmG W223A mutant, 0.2 mM PLP, and 2 mM amino donor (L-Gln, L-Glu, or L-Ala) at 30 °C. The reaction was analyzed with a CD spectrophotometer after blanked with Tris buffer. For CD spectrophotometer analysis, one milliliter quartz cuvettes were used, and spectra from 500 to 300 nm were recorded.

The CRM M conversion reactions were performed in 50 μL assay solution (50 mM Tris pH 7.5, 0.1 M NaCl, and 5% Glycerol) containing 4 μM CrmG, 80 μM

PLP, 0.3 mM amino donor (L-Glu, L-Gln, or L-Ala), 0.2 mM CRM M, at 30 °C for 0–48 h. HPLC was used to analyze CRM M consumption and CRM N production in the reaction. The HPLC analysis was carried out on a C18 column (250 × 4.6 mm, 5 μm) with a running program previously described[21]. Briefly, a gradient was run from 5% CH₃CN to 30% CH₃CN for 18 min, UV detection was performed at 313 nm.

The $K_i$ value of α-ketoglutarate or pyruvate toward CrmG was determined at 30 °C in 25 μL assay solution (200 mM Tris pH 7.5, 0.1 M NaCl, and 5% Glycerol) containing 4 μM CrmG, 80 μM PLP, 100 mM L-Glu, 0–5 mM CRM M, and 0/ 50 mM/100 mM α-ketoglutarate or pyruvate. Competitive inhibition constant of α-ketoglutarate or pyruvate toward CrmG was calculated by nonlinear regression analysis.

The kinetic parameters of CrmG toward amino donor L-Gln, L-Glu, and L-Ala were determined at 30 °C in 50 μL assay solution (200 mM Tris pH 7.5, 0.1 M NaCl, and 5% glycerol) containing 4 μM CrmG, 80 μM PLP, 0.2 mM CRM M and amino donor L-Gln/L-Glu/L-Ala.

**QM/MM study**. The crystal structures obtained in our previous work (PDB code: 5DDS and 5DDU) were employed as the initial models for the QM/MM simulation. A multi-step MM MD simulation performed with pmemd.cuda in AMBER12[37] was adopted to equilibrate the complex structure. The structure in the stable stage of the 50 ns MD simulations was chosen as the initial model for QM/ MM simulations. Periodic boundary condition was considered in MM MD, whereas spherical boundary condition was employed in QM/MM simulation. QM atoms were all described by DFT method (B3LYP) at 6–31G* basis set and MM atoms were described by the AMBER99SB force field[38]. The QM/MM boundary was treated by the improved pseudo bond approach and the 12 and 18 Å cutoffs were employed for van der Waals and electrostatic interactions, respectively. The QM/MM systems were minimized again for several iterations, and more than 5 ps QM/MM MD simulations were performed. The minimal time step of integration (1 fs/step) and equilibration time are adopted to give a more relaxed reaction system. The resulting conformations were used to map out the minimum energy path with the reaction coordinate driving method[39]. All the QM/MM simulations were run with QChem4.0[40] and Tinker[41] programs. The whole protocol is transplanted from our previous studies and has been carefully validated[42–44]. Additional

benchmark tests on the computational level (functionals and basis sets), choice of QM region and simulation time are described in Supporting Information (Supplementary Figs. 10–12).

**Reporting summary**. Further information on research design is available in the Nature Research Reporting Summary linked to this article.

## Data availability

The coordinates and diffraction data for the crystal structures were deposited in Protein Data Bank with the access codes of 6JC7, 6JC8, 6JC9, 6JCA, and 6JCB. Source data underlying plots shown in figures are provided in Supplementary Data 1.

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

## Acknowledgements

We thank the staff members of BL19U1 beamline at the National Center for Protein Science Shanghai and BL17U1 beamline at the Shanghai Synchrotron Radiation Facility, People's Republic of China, for assistance during the diffraction data collection. The work was financially supported by grants from National Key Research and Development Program Grant (2017YFA0504104), the National Science & Technology Major Project "Key New Drug Creation and Manufacturing Program", China (2018ZX09711002), National Natural Science Foundation of China (31500638, 31470204, 21903089), Guangdong Provincial Key Laboratory of Biocomputing (2016B030301007), Youth Innovation Promotion Association of the Chinese Academy of Sciences (2018390) to J.X., and Qingdao National Laboratory for Marine Science and Technology (QNLM2016ORP0304).

## Author contributions

J.X., Y. Zhu, Z.Y., K.S., Y. Zhang, and Y.D. performed experiments. J.L. and J.X. designed the experiments. J.L., J.X., Y. Zhang, W.Z., and C.Z. analyzed the data. X.T. and R.W. performed QM/MM study. J.L. and J.X. wrote the manuscript.

## Competing interests

The authors declare no competing interests.
