## [Peer Review File · Communications Biology]

Reviewers' comments:

Reviewer #1 (Remarks to the Author):

Strength:

- 1) The title properly justifies the subject of the paper.
- 2) The abstract provides an accessible summary of the paper.
- 3) The conclusion is consistent with the evidences and arguments presented.
- 4) All the figures and plots are clear.
- 5) Results are critical analysis of the data collected.

Comments:

1) A General schematic representation of the omega-transaminase catalysed reaction highlighting the role of cofactor and using amino donor should be provided for better understanding of the mechanism (Figure 7A)

2) The comparative pictures of the crystal structures of apo and bound complex should be given. (PDB ID- 5DDS, 5DDU)

3) The dynamics of catalytic pocket in apo CrmG, the structural rearrangement of catalytic pocket upon PLP binding, the main chain shift of catalytic base K344 between apo and active CrmG form, the loop dynamics should be validated from MD simulation analysis.

4) The ensemble structures for the roof region from MD simulation with initial structures of apo and bound conformations can be obtained for further validation.

5) In the mechanism (Figure 7A), the labeling of the structures like L-Glutamate, PLP, PMP, alpha-Ketoglutarate, CRM M, CRM N etc. should be mentioned.

6) Unfortunately, the QM-MM calculations are not validated with respect to functionals, basis set, choice of QM/MM region, time step of integration, equilibration time scale. These validations are necessary and should be provided in SI.

7) Wrong use of the half heded arrows in the mechanism. A few reaction arrows are not placed in proper directions (like the head of the arrow should be towards the less electron negative atom). Incorrect delocalization of electrons in some of the structures. (See for example the structures of A4, B1, B2, B3(**HOH), C2, D2, E1, E2, F1, F3)

On the basis of the above comments, I recommend a major revision of the manuscript.

Reviewer #2 (Remarks to the Author):

The authors present a well-written manuscript that describes a structural and mechanistic investigation of by-product inhibition in an omega-transaminase. This is achieved through x-ray crystallography, enzymatic assays and QM/MM. One of the main justifications of the work is that by-product inhibition is one of the major issues hindering application of these enzymes. Even though I am not sure I agree that this is as big of an issue as the authors think, the investigation does provide valuable insight into the mechanism of by-product inhibition and also some possible future solutions. I think the manuscript is generally of high quality but I do have some comments that need to be addressed:

1. Page 6, "Displaced unfavorable equilibrium of CrmG catalyzed CRM M amination" This part needs to be rewritten as there seems to be some confusion about the equilibrium. Equilibrium is a thermodynamic property and it is not affected by the catalyst. The results here are nice, but they only show that in this particular reaction, CRM M + amino donors, the equilibrium is shifted towards the product side. You cannot say that it is because the enzyme is not "suffering from unfavorable equilibrium" and you can definitely not compare it to another transaminase catalyzing another reaction as you do in the end of this part. The reductive amination of acetophenone is widely known to be thermodynamically challenging and the CrmG enzyme would also struggle with this reaction.

2. Page 9, line 210-211. "Although numerous structures of transaminase have been determined, this is the first report that a closed form catalytic pocket is identified." Are you sure about this statement? Is it not already shown in Humble et. al 2012 that the catalytic pocket goes into a closed conformation upon binding of PLP? Feel free to correct me if I am wrong. It could be that your definition of "closed" is different but then that needs to be clarified in the text.

3. Page 11, line 253-256. Here, you should cite: Schell et. al. Synthesis of pyridoxamine 5'-phosphate using an MBA:pyruvate transaminase as biocatalyst" (2009).

4. The language is generally ok but it needs to be generally improved throughout the text.

After a revision based on these comments, I would be happy to recommend acceptance of this manuscript.

Reviewer #3 (Remarks to the Author):

Xu et al. can show that absence of inhibition is due to weak binding of the products of the amine donors Gln, Glu and Ala. This is a very useful property of an omega-transaminase, not observed for other transaminase reactions. This is basically explained by weaker binding of oxoglutarate or pyruvate to CrmG compared to other transaminases.

Is this concept transferrable to other substrates and/or other transaminases? It should be made clearer, if only kinetics or also thermodynamics is responsible. In the latter case, CRM M transamination will remain a special case.

Since the enzyme cannot alter the thermodynamics of the Glu  oxoglutarate reaction, weaker binding of the product compared to other transaminases implies weaker binding of the substrates as well. The authors should report Km values for the amine donors and compare them with those of other transaminases. Large Km values would slow down the reaction unless large concentrations of the amine donors are used. How does CrmG compare to other transaminases in Km and kcat?

CRM M amination with Glu seems to have a favorable equilibrium constant (line 141-143). K must be at least as large (as favorable) as can be calculated from the 90% turnover. Is the conversion of acetophenone by transaminase ATA-103 limited by kinetics or by thermodynamics? In the latter case, the mechanism of CrmG will not help to get higher turnover, whatever mutagenesis might be applied to adapt CrmG to the substrate acetophenone.

Other comments (line numbers from the pdf file provided):

61 catalysis > catalysts (the catalysts are precious, but not the catalysis)

63 I would suggest: ... ω -transaminases have gained considerable attraction ...

121 write i as subscript in K_i

168 hereiafter > hereafter

216 Something is missing behind poly ((in "... containing poly")

260 I suggest "We should note that the PLP concentration is 10 times that of CrmG in these reactions."

267 Alanine > alanine

283 Use "gamma-carboxamido group" instead of "gamma-acyl amino group".

286 I would rather specify the OH group as phenolic OH-group than generally as hydroxyl group.

288 "form external aldimine" > "form the external aldimine".

296 carboxamido

311 "hydrolyzation" > "hydrolysis"

377 either "to synthesize amine compounds" or "in the synthesis of amine compounds" or "for synthesis of amine compounds".

378 transaminases > transaminases

383 circumvent > circumvents

383 facilitate > facilitates

401 and elsewhere: "°C" is generally missing missing in the pdf file

Fig. 5 The cofactor looks like the internal aldimine rather than PLP and Lys344 as described in the legend.

Reviewer #1

1) A General schematic representation of the omega-transaminase catalysed reaction highlighting the role of cofactor and using amino donor should be provided for better understanding of the mechanism (Figure 7A)

Answer: We thank reviewer for the suggestion. In Figure 1A, we have presented a general scheme of omega-transaminase catalyzed reaction. To further highlight the role of the cofactor, we have modified the Figure 1A, shown here:

Figure 1A

2) The comparative pictures of the crystal structures of apo and bound complex should be given.(PDB ID- 5DDS, 5DDU)

Answer: We thank reviewer for this suggestion. In the revision, we compared the crystal structures of apo and PLP-bound CrmG and added Figure S2.

Figure S2. Structures of apo CrmG are superposed on PLP-bound CrmG (PDB code 5DDS). Structures of apo CrmG in C2 space group and I222 space group are colored in yellow and green, respectively; Structure of PLP-bound CrmG is colored in blue, internal aldimine is shown as white stick.

3) The dynamics of catalytic pocket in apo CrmG, the structural rearrangement of catalytic pocket upon PLP binding, the main chain shift of catalytic base K344 between apo and active CrmG form, the loop dynamics should be validated from MD simulation analysis.

Answer: We thank reviewer for the suggestion. We performed additional MD simulation. In the revised manuscript, structures of some key segments of CrmG from the MD simulations with different conformations are displayed in Figure S3 and S5. Structural difference of the “roof” in active form and apo form is validated by MD simulations, and RMSF data validates the flexibility of the “roof” in apo CrmG of C2 space group. The result that segment S206-V212 shows as a helix in closed apo form and as a loop in both open apo form and active form is also validated. Structure analysis and key distances tracking along the MD trajectory also confirm the position shift of Q318, as the distance of side chains in V317 and Q318 is more fluctuant in the apo form. Moreover, distance of V317 and K344 is longer and more fluctuant in apo form, which is in agreement with the main chain shift of catalytic base K344 in apo and active form CrmG observed in the crystal structures.

Figure S3. The ensemble structures for region “roof” of CrmG from MD simulations with active form (A), closed apo form (B) and open apo form (C). The two monomers are color by light green and gray, segments of S206-V212 and G213-S232 are color in blue and magenta, K344 and PLP

are shown as orange stick. (D) RMSF of CrmG in active, open and closed form. The “roof” region is shown in expanded view.

Figure S5. (A) Comparison of β_{10}/α_{10} loop and K344 on apo CrmG (pink) with that on PLP-Bound CrmG (light green) from MD simulations, and key distances tracking along the MD trajectory. V317, Q318 K344 and PLP are shown as stick, some key atoms that used to track the segment movement are also labeled. (B) Distance of $C\alpha-C\beta$ (black line), $N-C\beta$ (red line), represent the main chain and side chain shift of K344, and $C-C\beta$ (blue line) represents the positional shift of Q318. The solid lines represent the distance changes in active form of CrmG and dashed lines represent those in apo form.

4) The ensemble structures for the roof region from MD simulation with initial structures of apo and bound conformations can be obtained for further validation.

Answer: In the revised manuscript, we performed MD simulation for the roof region of apo and bound conformations (Figure S3).

5) In the mechanism (Figure 7A), the labeling of the structures like L-Glutamate, PLP, PMP, alpha-Ketoglutarate, CRM M, CRM N etc. should be mentioned.

Answer: We thank reviewer for the suggestion. We the picture in question is Figure 6A, not Figure 7A. In the revised manuscript, L-Glutamate, PLP, PMP, alpha-Ketoglutarate, CRM M, and CRM N are labeled in Figure 6A.

Figure 6A Detailed reaction mechanism of CrmG catalyzed transamination. Cofactors PLP and PMP are colored in orange; amino donor L-Glu, amino acceptor CRM M and by-product α -ketoglutarate (α -KG) are colored in blue.

6) Unfortunately, the QM-MM calculations are not validated with respect to functionals, basis set, choice of QM/MM region, time step of integration, equilibration time scale. These validations are necessary and should be provided in SI.

Answer: Thank you for the suggestions. Benchmark tests on the computational level (functionals and basis sets), choice of QM region and simulation time scale are added in Supporting Information (Figure S10-S12). All the results confirm the reliability of the protocol used in our current QM/MM simulations on CrmG. For the time step of integration, the minimal parameter (1 fs/step) is used in our current work, which is transplanted from our previous studies and has been carefully validated (ACS Catal. 2020, 10, 1470-1484., Angew. Chem. Int. Ed. 2015, 54, 8693-8696., ACS Catal. 2016, 6, 6918-6929.).

7) Wrong use of the half heded arrows in the mechanism. A few reaction arrows are not placed in proper directions (like the head of the arrow should be towards the less electron negative atom). Incorrect delocalization of electrons in some of the structures. (See for examplded the structures of A4, B1,B2,B3(**HOH), C2,D2,E1,E2,F1,F3)

Answer: We apologize for the mistake. We have fixed in the revised manuscript (Figure 6A).

Reviewer #2

1. Page 6, "Displaced unfavorable equilibrium of CrmG catalyzed CRM M amination" This part needs to be rewritten as there seems to be some confusion about the equilibrium. Equilibrium is a thermodynamic property and it is not affected by the catalyst. The results here are nice, but they only show that in this particular reaction, CRM M + amino donors, the equilibrium is shifted towards the product side. You cannot say that it is because the enzyme is not "suffering from unfavorable equilibrium" and you can definitely not compare it to another transaminase catalyzing another reaction as you do in the end of this part. The reductive amination of acetophenone is widely known to be thermodynamically challenging and the CrmG enzyme would also struggle with this reaction.

Answer: We apologize for our mistake and thanks for the comments. In the revised manuscript,

this part was rewritten as "Favorable equilibrium of CRM M conversion **by CrmG**"

Besides by-product inhibition, unfavorable equilibrium caused by by-product is another fundamental hurdle for the widespread application of transaminase. To determine whether CRM M conversion suffers from unfavorable equilibrium, we investigate the equilibrium of CrmG catalyzed CRM M conversion. We previously reported that, without by-product removal in situ, biaryl aldehyde CRM M can be completely converted to biaryl amine CRM N by CrmG using high excess (50 equiv) of L-Glu, L-Gln, or L-Ala as the amino donor (Table S2). Here, we carried out reactions of CrmG catalyzing CRM M amination with 1.5 equiv amino donor L-Glu, L-Gln, or L-Ala (Figure 1D). Our results showed that, without by-product removal in situ, about 90% CRM M can be converted to CRM N with amino donor L-Glu and L-Gln, in 12h and 48h, respectively. With amino donor L-Ala, 50% conversion of CRM M to CRM N was achieved in 48h. This indicating high conversion of CRM can be achieved with little excess amino donor L-Glu, L-Gln, or L-Ala."

2. Page 9, line 210-211. "Although numerous structures of transaminase have been determined, this is the first report that a closed form catalytic pocket is identified." Are you sure about this statement? Is it not already shown in Humble et. al 2012 that the catalytic pocket goes into a closed conformation upon binding of PLP? Feel free to correct me if I am wrong. It could be that your definition of "closed" is different but then that needs to be clarified in the text.

Answer: We thank reviewer for the comments. The active site of *Chromobacterium violaceum* (Cv- ω TA) was reorganized and transited from inactive form to active form upon PLP binding. The active site of PLP-bound Cv- ω TA was accessible by substrates. However, in the closed form catalytic pocket of apo CrmG, the active base was totally blocked, and inaccessible by cofactor and substrates.

To make it more clear, this sentence is rewritten as:

"To our knowledge, although numerous structures of ω -transaminase have been determined, here,

we firstly report a closed form catalytic pocket of apo ω -transaminase that the active base was inaccessible to the solvent."

3. Page 11, line 253-256. Here, you should cite: Schell et. al. Synthesis of pyridoxamine 5'-phosphate using an MBA:pyruvate transaminase as biocatalyst" (2009).

Answer: We thank reviewer for the suggestion. This reference was cited in the revised manuscript.

4. The language is generally ok but it needs to be generally improved throughout the text.

Answer: We thank reviewer for the suggestion. The language was improved in the revised manuscript.

Reviewer #3

(1) Is this concept transferrable to other substrates and/or other transaminases? It should be made clearer, if only kinetics or also thermodynamics is responsible. In the latter case, CRM M transamination will remain a special case.

Answer: We thank reviewer for the comments. We performed the reaction of CrmG catalyzed acetophenone conversion. Unfortunately, we couldn't detect any reaction. It may indicate that acetophenone cannot be recognized by CrmG. However, we believe that the proposed by-product inhibition overcoming mechanism has potential to be transferrable to some other substrates and/or transaminases via protein engineering. Inhibition is a kinetic property. Our proposed mechanism for CrmG overcoming by-product inhibition is that the flexible roof harms by-product binding to PMP-bound CrmG.

(2) Since the enzyme cannot alter the thermodynamics of the Glu \rightarrow oxoglutarate reaction, weaker binding of the product compared to other transaminases implies weaker binding of the substrates as well. The authors should report K_m values for the amine donors and compare them with those of other transaminases. Large K_m values would slow down the reaction unless large concentrations of the amine donors are used. How does CrmG compare to other transaminases in K_m and k_{cat} ?

Answer: We thank reviewer for the suggestion. In the revised manuscript, we determined the K_m and K_{cat} for CrmG toward L-Glu, L-Gln and L-Ala (Figure 5D). The binding of L-Glu, L-Gln and L-Ala to CrmG is significantly stronger than isopropylamine binding to OA- ω TA ($K_m=4.7 \times 10^5 \mu\text{M}$) (Adv. Synth. Catal. 2015, 357, 1732 – 1740).

Figure 5D. Determination kinetic parameters of CrmG toward amino donor L-Gln, L-Glu or L-Ala.

(3) CRM M amination with Glu seems to have a favorable equilibrium constant (line 141-143). K must be at least as large (as favorable) as can be calculated from the 90% turnover. Is the conversion of acetophenone by transaminase ATA-103 limited by kinetics or by thermodynamics? In the latter case, the mechanism of CrmG will not help to get higher turnover, whatever mutagenesis might be applied to adapt CrmG to the substrate acetophenone.

Answer: We thank reviewer for the comments. As CrmG did not convert acetophenone in our experiment, we cannot figure out whether conversion of acetophenone by transaminase ATA-103 is limited by kinetics or by thermodynamics. Based on current knowledge, we agree with the comments given by reviewer2 that equilibrium is a thermodynamic property and is not affected by the catalyst.

Other comments (line numbers from the pdf file provided):

61 catalysis > catalysts (the catalysts are precious, but not the catalysis)

Answer: We apologize for the mistake and appreciate the reviewer for the comments. We have fixed it in the revised manuscript.

63 I would suggest: ... ω -transaminases have gained considerable attraction ...

Answer: We have fixed it in the revised manuscript.

121 write i as subscript in K_i

Answer: We have fixed it in the revised manuscript.

168 hereinafter > hereafter

Answer: We have fixed it in the revised manuscript.

216 Something is missing behind poly () in "... containing poly")

Answer: We have fixed it in the revised manuscript. It should read "Poly(acrylic acid)".

260 I suggest "We should note that the PLP concentration is 10 times that of CrmG in these reactions."

Answer: We have fixed it in the revised manuscript.

267 Alanine > alanine

Answer: We have fixed it in the revised manuscript.

283 Use "gamma-carboxamido group" instead of "gamma-acyl amino group".

Answer: We have fixed it in the revised manuscript.

286 I would rather specify the OH group as phenolic OH-group than generally as hydroxyl group.

Answer: We have fixed it in the revised manuscript.

288 "form external aldimine" > "form the external aldimine".

Answer: We have fixed it in the revised manuscript.

296 carboxamido

Answer: We have fixed it in the revised manuscript.

311 "hydrolyzation" > "hydrolysis"

Answer: We have fixed it in the revised manuscript.

377 either "to synthesize amine compounds" or "in the synthesis of amine compounds" or "for synthesis of amine compounds".

Answer: We have fixed it in the revised manuscript.

378 transaminases > transaminases

Answer: We have fixed it in the revised manuscript.

383 circumvent > circumvents

Answer: We have fixed it in the revised manuscript.

383 facilitate > facilitates

Answer: We have fixed it in the revised manuscript.

401 and elsewhere: "°C" is generally missing missing in the pdf file

Answer: °C does exist in our Word document, we will make sure this shows up correctly in the

PDF file.

Fig. 5 The cofactor looks like the internal aldimine rather than PLP and Lys344 as described in the legend.

Answer: We have fixed it in the revised manuscript.

REVIEWERS' COMMENTS:

Reviewer #1 (Remarks to the Author):

Thank you for addressing all my concerns satisfactorily.

Reviewer #2 (Remarks to the Author):

The authors have provided satisfactory answers and revisions to all reviewer comments. I am now happy to recommend acceptance of this manuscript in Nature Communications Biology.

Reviewer #3 (Remarks to the Author):

The authors now give a complete example of a transaminase reaction that is not inhibited by the product. The claim that this "facilitates the rational design of transaminase to eliminate by-product inhibition" (line 377) is not yet proven to me and this limitation should be mentioned. As a useful basis for further experiments, though, I support publication of the manuscript.

To me, it is still not fully clear, which contribution kinetics and thermodynamics have. This is on the academic side, though, it might not bother the biotechnologist. The real technological benefit of the proposed mechanism will show, if in case of an existing enzyme catalyzed substrate conversion, which is product inhibited, using CrmG or an accordingly mutated transaminase relieves the product inhibition. This is a separate project and beyond the scope of this paper. The fact of this limitation (that the same reaction has not been compared yet using CrmG without by-product inhibition on one hand and using a different transaminase with by-product inhibition on the other hand) should be mentioned in the discussion with the outlook and goal to analyze this in further research.

Fig. 5 D

Is there a reason to use upper case "K" for k_{cat} ? Otherwise, I would recommend lower case "k".

Comparison of $K_m(\text{Glu})$ for CrmG to $K_m(\text{isopropylamine})$ for OA-omegaTA is only half the story. Do you have K_m values for the same substrates with CrmG vs. another transaminase and "no product inhibition" vs. "product inhibition"?

The K_m for glutamate is low enough, though, such that low mM glutamate concentrations will saturate the enzyme. This is technologically useful.

(1) To me, it is still not fully clear, which contribution kinetics and thermodynamics have. This is on the academic side, though, it might not bother the biotechnologist. The real technological benefit of the proposed mechanism will show, if in case of an existing enzyme catalyzed substrate conversion, which is product inhibited, using CrmG or an accordingly mutated transaminase relieves the product inhibition. This is a separate project and beyond the scope of this paper. The fact of this limitation (that the same reaction has not been compared yet using CrmG without by-product inhibition on one hand and using a different transaminase with by-product inhibition on the other hand) should be mentioned in the discussion with the outlook and goal to analyze this in further research.

Answer: We thank the reviewer for the suggestion. In the revised manuscript, we have added the following in the conclusion section: "Further studies, especially on protein engineering, will need to be carried out to validate our proposed mechanism of overcoming by-product inhibition of ω -transaminases."

(2) Fig. 5 D. Is there a reason to use upper case "K" for kcat? Otherwise, I would recommend lower case "k".

Answer: We have fixed it in the revised manuscript.

(3) Comparison of $K_m(\text{Glu})$ for CrmG to $K_m(\text{isopropylamine})$ for OA- ω TA is only half the story. Do you have K_m values for the same substrates with CrmG vs. another transaminase and "no product inhibition" vs. "product inhibition"?

The K_m for glutamate is low enough, though, such that low mM glutamate concentrations will saturate the enzyme. This is technologically useful.

Answer: We thank the reviewer for this comment. From the literature, there is no report on the K_m values of the amino donor Glu or Ala for transaminase with severe by-product inhibition (such as OA- ω TA).